# 5-Oxo-hexahydroquinoline Derivatives and Their Tetrahydroquinoline Counterparts as Multidrug Resistance Reversal Agents

**DOI:** 10.3390/molecules25081839

**Published:** 2020-04-16

**Authors:** Omolbanin Shahraki, Mehdi Khoshneviszadeh, Mojtaba Dehghani, Maryam Mohabbati, Marjan Tavakkoli, Luciano Saso, Najmeh Edraki, Omidreza Firuzi

**Affiliations:** 1Cellular and Molecular Research Center, Resistant Tuberculosis Institute, Zahedan University of Medical Sciences, Zahedan 98167-43463, Iran; o.shahraki@gmail.com; 2Medicinal and Natural Products Chemistry Research Center, Shiraz University of Medical Sciences, Shiraz 71348-53734, Iran; m.khoshneviszadeh@gmail.com (M.K.); Dehghanimojtaba58@yahoo.com (M.D.); m_mohabati2000@yahoo.com (M.M.); tavakkoli_marjan@yahoo.com (M.T.); 3Department of Medicinal Chemistry, Faculty of Pharmacy, Shiraz University of Medical Sciences, Shiraz 71468-64685, Iran; 4Department of Physiology and Pharmacology “Vittorio Erspamer”, Sapienza University of Rome, P. le Aldo Moro 5, 00185 Rome, Italy; luciano.saso@uniroma1.it

**Keywords:** anticancer drug resistance, efflux pumps, antiproliferative agents, drug design, 1,4-dihydropyridine

## Abstract

Cancer is a leading cause of death worldwide. Multidrug resistance (MDR) is a main reason of chemotherapy failure in many patients and is often related to overexpression of ATP-binding cassette (ABC) transporters, including P-glycoprotein (P-gp/ABCB1). Agents that are capable of modulation of the activity of these transporters might be effective in overcoming MDR. In this study, a new set of 1,4,5,6,7,8-hexahydro 5-oxo quinoline-3-carboxamide derivatives bearing 4-methylthiazole moiety and their tetrahydroquinoline counterparts were synthesized. MDR reversal activity of these 16 newly synthesized derivatives was tested in P-gp overexpressing MES-SA-DX5 human uterine sarcoma cells by flow cytometric determination of Rhodamine123 efflux. The effect of the most potent compounds in induction of apoptosis and alterations of cell cycle was examined in these cells by a flow cytometric method. Inherent cytotoxicity of the synthesized compounds was evaluated against MCF-7, A-549 and K562 cancer cell lines, as well as MES-SA-DX5 and their parental non-resistant MES-SA and also HEK-293 non-cancerous cells by MTT assay. Compounds **A1** and **A2** with 5-oxo-hexahydroquinoline structure bearing 2,4-dichlorophenyl and 4-bromophenyl moieties, respectively, and their tetrahydroquinoline counterparts **B1** and **B2** significantly blocked P-gp efflux, induced apoptosis and showed the highest cytotoxicities against MES-SA-DX5 cells. However, only **A2** and **B2** compounds were relatively selective against cancer and MDR cells as compared to non-resistant and non-cancerous cells. These findings demonstrate that 5-oxo-hexahydroquinoline and 5-oxo-tetrahydroquinoline derivatives represent promising agents with therapeutic potential in drug resistant cancers.

## 1. Introduction

Cancer is a major cause of death and disability worldwide and despite considerable diagnostic and therapeutic advancements in recent years, it is still the cause of more than 8 million deaths in the world every year [1]. The resistance of malignant tumor cells to multiple structurally and mechanistically unrelated classes of anticancer agents is recognized as multidrug resistance (MDR). This phenomenon very often occurs in advanced cancer cases and is a major cause of failure of therapies [2,3]. Different mechanisms are involved in MDR: Very often it is caused by the overexpression of P-glycoprotein (P-gp), also named multidrug resistance protein-1 (MDR1), which is encoded by ATP-binding cassette (ABC) subfamily B member 1 (ABCB1) gene [4]. P-gp has been one of the first members of ABC transporters to be studied [5]. It functions as an efflux pump extruding several substrate molecules and plays an important physiological role in cleaning cells from xenobiotics and maintaining the integrity of blood-brain barrier [6]. However, overexpression of P-gp in cancer cells leads to reduced accumulation of cytotoxic drugs and even targeted therapeutics resulting in resistance against these agents [3,7].

In this context, targeting P-gp by small-molecule inhibitors seems to be a promising approach for overcoming MDR and restoring chemosensitivity in tumor cells [4]. In spite of numerous considerable efforts using different classes of compounds, there is no approved MDR-targeting drug available yet [3,8,9]. Hence, new P-gp inhibitors with high potency and selectivity are much needed to be developed.

One important class of candidate compounds capable of MDR reversal, is 1,4-dihydropyridine derivatives. 1,4-DHPs possess several important pharmacological properties including calcium-channel-blocking, bronchodilator, anti-ischemic and antitumor as well as MDR reversal effects [10,11,12,13,14,15]. Several agents structurally related to 1,4-DHP derivatives have been reported as P-gp inhibitors [16,17,18,19,20,21].

We have recently designed and synthesized a set of 5-oxohexahydroquinoline [22] and tetrahydroquinolinone derivatives [20] closely related structures to 1,4-DHPs, as MDR reversal agents. Here, in another attempt to respond to the crucial need for novel cancer therapeutic agents and as part of our research program toward the discovery of useful MDR reversal compounds, we report the synthesis and biological evaluation a set of sixteen 5-oxo-hexahydroquinoline derivatives. The findings of the study showed that some of these compounds are capable of reversing MDR in P-gp overexpressing cancer cells.

## 2. Materials and Methods

### 2.1. Chemistry

All the reagents were bought from Sigma Aldrich chemical Co. or Merck and used without further purification. Melting points were recorded by a hot stage apparatus (Electrothermal, Essex, UK) and reported without correction. ^1^H-NMR spectra were determined by a Bruker FT-400 MHz spectrometer (Bruker Daltonics, Bremen, Germany) in DMSO-d6. All the chemical shifts were reported as (d) values (ppm) against tetramethylsilane as an internal standard. The MS spectra were recorded using an Agilent 7890A spectrometer at 70 eV. Elemental analyses were performed on a TESCAN-Vega3 CNO analyzer (TESCAN, Brno, Czech Republic).

#### General Procedure of the Synthesis

2-Methylthiazol-4-amine (5 mmol) was mixed with 2,2,6-trimethyl-4*H*-1,3-dioxin-4-one (6 mmol) in 10 mL xylene. The mixture was refluxed for 2–4 h. The reaction procedure was monitored using TLC (thin layer chromatography). At the end of reaction, the precipitate was removed and washed with petroleum ether (30–50 mL). The obtained product, *N*-(2-methylthiazol-4-yl)-3-oxobutanamide, cyclohexane-1,3-dione and different aryl aldehydes were mixed in equimolar and excess amount of ammonium acetate was added. The mixture was refluxed in ethanol for 24 h. The oxidized products were achieved with high purity via oxidation for 24–48 h in the presence of MnO_2_ and ethanol as solvent. The acquired products of each step were characterized using ^1^H-NMR, ^13^C-NMR, mass and IR spectroscopy. All spectra are shown in Appendix A.

*N-(2-Methylthiazol-4-yl)-3-oxobutanamide* (**a**)**:** Yield: 86%, yellow powder, mp: 173–175 °C, ^1^H-NMR (DMSO-d6) δ: 11.65 (1H, brs, amide N-H), 7.26 (1H, Apparent s, thiazole C_5_-H), 3.76 (2H, s, COCH_2_CO), 2.35 (3H, s, CH_3_CO), 2.21 (3H, Apparent, thiazole-CH_3_); IR (KBr) ν (cm^−1^): 3179 (N-H, amide), 3053 (C-H, aromatic), 2974, 2899 (C-H, aliphatic), 1723 (C=O, ketone), 1672 (C=O, amide); MS *m*/*z* (%): 198(29) [M+], 141(8), 114(100), 72(23), 43(65).

*4-(2,4-Dichlorophenyl)-2-methyl-N-(2-methylthiazol-4-yl)-5-oxo-1,4,5,6,7,8-hexahydroquinoline-3-carboxamide (***A1**)**:** Yield 95%; yellow powder; mp. 195 °C; ^1^H-NMR (500MHz, DMSO-d_6_) δ: 11.92 (s, 1H, amide N-H), 8.95 (s, 1H, NH-DHP), 7.33 (d, 1H, *J* = 2.0 Hz, C_3_-H_)_, 7.30 (d, 1H, *J* = 8.5 Hz, C_5_-H), 7.23 (d, 1H, *J* = 8.5 Hz, C_6_-H), 6.67 (s, 1H, thiazole-H), 5.35 (s, 1H, DHP C_4_-H), 2.23 (s, 3H, thiazole-CH_3_), 2.21-2.09 (m, 2H, cyclohexanone C_6_-H), 1.93 (s, 3H, DHP-CH_3_), 1.91-1.79 (m, 2H, cyclohexanone C_8_-H), 1.82-1.80 (m, 2H, cyclohexanone C_7_-H). ^13^C-NMR (125MHz, DMSO-d_6_): 194.19, 153.37, 144.15, 136.18, 132.67, 132.61. 131.38, 128.45, 127.89, 109.25, 108.61. 107.78, 37.18, 36.03, 26.84, 25.01, 21.52, 21.36, 19.02, 17.36, 17.12. IR (KBr) ν (cm^−1^): 3197 (DHP N-H), 3080 (C-H, aromatic), 2947 (C-H, aliphatic), 1675, 1608 (C=O, amide, ketone); MS *m*/*z* (%): 449 (30) [M + 2]^+^, 448 (8) [M + 1]^+^, 447 (22) [M], 412 (19), 334 (100), 298 (16), 272 (28), 162 (35), 115 (29), Anal. Calcd: C, 56.26; N, 9.37; O, 7.14. Found: C, 56.19; N, 9.25; O, 7.09.

*4-(4-Bromophenyl)-2-methyl-N-(2-methylthiazol-4-yl)-5-oxo-1,4,5,6,7,8-hexahydroquinoline-3-carboxamide* (**A2**): Yield 92%; yellow powder; mp. 273 °C; ^1^H-NMR (500MHz, DMSO-*d*_6_) δ: 11.80 (s, 1H, amide N-H), 8.97 (s, 1H, NH-DHP), 7.38 (d, 2H, *J* = 8.5 Hz, C_3_-H_,_ C_5_-H_)_, 7.13 (d, 2H, *J* = 8.5 Hz, C_2_- H_,_ C_6_-H), 6.66 (s, 1H, thiazole-H), 5.08 (s, 1H, DHP C_4_-H), 2.48-2.46 (m, 2H, cyclohexanone C_8_-H), 2.24 (s, 3H, thiazole-CH_3_), 2.20–2.14 (m, 2H, cyclohexanone C_6_-H), 2.10 (s, 3H, DHP-CH_3_), 1.92–1.72 (2 × m, 2H, cyclohexanone C_7_-H). ^13^C-NMR (125MHz, DMSO-d_6_): 194.59, 152.46, 146.72, 138.75, 131.30, 130.19, 119.37, 109.89, 107.92, 107.88, 107.83, 56.49, 37.40, 37.20, 26.75, 21.24, 19.02, 17.72, 17.30. IR (KBr) ν (cm^−1^): 3191 (DHP N-H), 3055 (C-H, aromatic), 2945 (C-H, aliphatic), 1667, 1607 (C=O, amide, ketone) ; MS *m*/*z* (%): 459 (34) [M + 2]^+^, 457 (34) [M], 346 (74), 344 (80), 317 (34), 300 (26), 264 (19), 162 (100), Anal. Calcd: C, 55.03; N, 9.17; O, 6.98. Found: C, 54.99; N, 9.03; O, 6.73.

*4-(3-Chlorophenyl)-2-methyl-N-(2-methylthiazol-4-yl)-5-oxo-1,4,5,6,7,8-hexahydroquinoline-3-carboxamide* (**A3**): Yield 88%; yellow powder; mp. 253 °C; ^1^H-NMR (500MHz, DMSO-*d*_6_) δ: 11.83 (s, 1H, amide N-H), 9.00 (s, 1H, NH-DHP), 7.22 (t, 1H, *J* = 8.0 Hz, C_4_-H), 7.18 (s, 1H, C_2_-H), 7.14 (t, 2H, *J* = 6.0 Hz, C_5_-H, C_6_-H), 6.67 (s, 1H, thiazole-H), 5.12 (s, 1H, DHP C_4_-H), 2.48–2.47 (m, 2H, cyclohexanone C_8_-H), 2.23 (s, 3H, thiazole-CH_3_), 2.21–2.17 (m, 2H, cyclohexanone C_6_-H), 2.13 (s, 3H, DHP-CH_3_), 1.92-1.74 (2 × m, 2H, cyclohexanone C_7_-H). ^13^C-NMR (125MHz, DMSO-*d*_6_): 194.65, 152.63, 149.69, 139.02, 139.00, 137.63, 133.10, 130.32, 127.74, 126.69, 126.30, 109.73, 107.70, 56.49, 37.65, 37.17, 26.75, 21.22, 19.01, 17.77, 17.27. IR (KBr) ν (cm^−1^): 3189 (DHP N-H), 3065 (C-H, aromatic), 2970 (C-H, aliphatic), 1670, 1604 (C=O, amide, ketone), 1548, 1378 (C-NO_2_ aromatic); MS *m*/*z* (%): 415 (21) [M + 2]^+^, 413 (51) [M], 302 (47), 300 (100), 264 (36), 115 (37), Anal. Calcd: C, 60.94; N, 10.15; O, 7.73. Found: C, 60.82; N, 10.08; O, 7.46.

*2-Methyl-N-(2-methylthiazol-4-yl)-5-oxo-4-(3,4,5-trimethoxyphenyl)-1,4,5,6,7,8-hexahydroquinoline-3-carboxamide* (**A4**): Yield 81%; yellow powder; mp. 160 °C; ^1^H-NMR (500MHz, DMSO-d_6_) δ: 8.99 (s, 1H, NH-DHP), 6.67 (s, 1H, thiazole-H), 6.41 (s, 2H, C_2_-H, C_6_-H), 5.09 (s, 1H, DHP C_4_-H), 3.62 (s, 6H, -OCH_3_), 3.57 (s, 3H, -OCH_3_), 2.56–2.51(m, 3H, thiazole-CH_3_), 2.26–2.23 (m, 5H, cyclohexanone C_6_-H, cyclohexanone C_8_-H), 2.09 (s, 3H, DHP-CH_3_), 1.96–1.81 (2 × m, 2H, cyclohexanone C_7_-H).^13^C-NMR (125MHz, DMSO-*d*_6_): 194.77, 167.75, 152.91, 162.67, 146.76, 143.03, 138.06, 136.05, 109.52, 108.65, 107.67, 104.78, 60.25, 56.48, 55.95, 37.74, 37.34, 26.83, 21.42, 19.01, 17.67, 17.30. IR (KBr) ν (cm^−1^): 3305 (DHP N-H), 2932 (C-H, aromatic), 2895 (C-H, aliphatic), 1609 (C=O, amide, ketone), 1267, 1002 (C-O, aromatic); MS *m*/*z* (%): 469 (19) [M]^+^, 356 (34), 324 (20), 300 (15), 218 (16), 170 (18), 156 (100), Anal. Calcd: C, 61.39, N, 8.95; O, 17.04. Found: C, 61.28; N, 8.75; O, 16.94.

*4-(4-Chlorophenyl)-2-methyl-N-(2-methylthiazol-4-yl)-5-oxo-1,4,5,6,7,8-hexahydroquinoline-3-carboxamide* (**A5**)**:** Yield 90%; yellow powder; mp. 253 °C; ^1^H-NMR (500MHz, DMSO-*d*_6_) δ: 11.79 (s, 1H, amide N-H), 8.97 (s, 1H, NH-DHP), 7.24 (d, 2H, *J* = 8.0 Hz, C_3_-H_,_ C_5_-H), 7.18 (d, 2H, *J* = 8.0 Hz, C_2_- , C_6_- H), 6.67 (s, 1H, thiazole-H), 5.09 (s, 1H, DHP C_4_-H), 2.48–2.46(m, 2H, cyclohexanone C_8_-H), 2.23 (s, 3H, thiazole-CH_3_), 2.23–2.14 (m, 2H, cyclohexanone C_8_-H), 2.10 (s, 3H, DHP-CH_3_), 1.92–1.75 (2 × m, 2H, cyclohexanone C_7_-H). ^13^C-NMR (125MHz, DMSO-*d*_6_): 194.68, 172.20, 152.45, 139.28, 130.82, 129.76, 128.38, 123.93, 109.94, 43.70, 43.68, 42.70, 38.94, 37.20, 26.75, 21.24, 17.72, 17.05. IR (KBr) ν (cm^−1^): 3269 (DHP N-H), 3195 (C-H, aromatic), 2956 (C-H, aliphatic), 1667, 1608 (C=O, amide, ketone); MS *m*/*z* (%): 415 (10) [M + 2]^+^, 413 (3) [M]^+^, 300 (44), 298 (100), 267 (30), 239 (39), Anal. Calcd: C, 60.94; N, 10.15; O, 7.73. Found: C, 60.86; N, 10.04; O, 7.41.

*4-(3-Ethoxy-4-hydroxyphenyl)-2-methyl-N-(2-methylthiazol-4-yl)-5-oxo-1,4,5,6,7,8-hexahydroquinoline-3-carboxamide* (**A6**): Yield 85%; yellow powder; mp. 251 °C; ^1^H-NMR (500MHz, DMSO-*d*_6_) δ: 11.63 (s, 1H, amide N-H), 8.85 (s, 1H, NH-DHP), 8.56 (s, 1H, OH), 6.71 (s, 1H, C_2_- H ), 6.62 (s, 1H, thiazole-H), 6.59 (d, 1H, *J* = 8.0 Hz, C_5_- H ), 6.52 (d, 1H, *J* = 8.0 Hz, C_6_-H), 4.98 (s, 1H, DHP C_4_-H), 3.91–3.76 (m, 2H, CH_3_-CH_2_-O), 2.48-2.45 (m, 2H, cyclohexanone C_8_-H), 2.24–2.19 (m, 5H, cyclohexanone C_6_-H, thiazole-CH_3_), 2.09 (s, 3H, DHP-CH_3_), 1.93–1.78 (2 × m, 2H, cyclohexanone C_7_-H). 1.23 (t, 3H, *J* = 7.0, CH_3_-CH_2_-O). ^13^C-NMR (125MHz, DMSO-*d*_6_): 194.66, 167.45, 155.99, 151.91, 146.97, 146.38, 145.36, 138.58, 138.29, 119.98, 115.64, 113.70, 110.50, 108.62, 107.73, 64.04, 37.34, 36.97, 26.79, 21.35, 17.70, 17.37, 15.12. (IR (KBr) ν (cm^−1^): 3282 (DHP N-H), 3067 (C-H, aromatic), 2990 (C-H, aliphatic), 1666, 1614 (C=O, amide, ketone); MS *m*/*z* (%): 439 (1) [M]^+^, 310 (10), 285 (57), 268 (14), 162 (100), Anal. Calcd: C, 62.85; N, 9.56; O, 14.56. Found: C, 62.45; N, 9.32; O, 14.18.

*4-(4-Methoxyphenyl)-2-methyl-N-(2-methylthiazol-4-yl)-5-oxo-1,4,5,6,7,8-hexahydroquinoline-3-carboxamide* (**A7**)**:** Yield 78%; yellow powder; mp. 185 °C; ^1^H-NMR (500MHz, DMSO-d_6_) δ: 11.82 (s, 1H, amide N-H), 8.89 (s, 1H, NH-DHP), 7.08 (d, 2H, *J* = 8.0 Hz, C_5_-H, C_3_-H), 6.74 (d, 2H, *J* = 8.0 Hz, C_2_- H, C_6_- H), 6.66 (s, 1H, thiazole-H), 5.04 (s, 1H, DHP C_4_-H), 3.66 (s, 3H, CH_3_-O), 2.47–2.44 (m, 2H, cyclohexanone C_8_-H), 2.23–2.17 (m, 5H, cyclohexanone C_6_-H, thiazole-CH_3_), 2.10 (s, 3H, DHP-CH_3_), 1.91–1.72 (2 × m, 2H, cyclohexanone C_7_-H). ^13^C-NMR (125MHz, DMSO-*d*_6_): 194.56, 172.48, 157.83, 151.89, 139.72, 139.67, 138.54, 128.88, 113.81, 110.68, 108.51, 107.69, 55.31, 37.28, 36.74, 26.76, 21.52, 21.29, 17.71, 17.31. (IR (KBr) ν (cm^−1^): 3196 (DHP N-H), 3076 (C-H, aromatic), 2943 (C-H, aliphatic), 1675, 1608 (C=O, amide, ketone); MS *m*/*z* (%): 407 (15) [M − 2]^+^, 294 (100), 251 (5), 223 (4), Anal. Calcd: C, 64.53; N, 10.26; O, 11.72. Found: C, 64.41; N, 10.20; O, 11.58.

*4-(3,4-Dimethoxyphenyl)-2-methyl-N-(2-methylthiazol-4-yl)-5-oxo-1,4,5,6,7,8-hexahydroquinoline-3-carboxamide* (**A8**)**:** Yield 73%; yellow powder; mp. 222 °C; ^1^H-NMR (500MHz, DMSO-*d*_6_) δ: 11.69 (s, 1H, amide N-H), 8.88 (s, 1H, NH-DHP), 6.76 (d, 2H, *J* = 8.0 Hz, C_5_-H , thiazole-H), 6.66–6.64 (m, 2H, C_2_-H , C_6_-H ), 5.05 (s, 1H, DHP C_4_-H), 3.65 (s, 3H, CH_3_-O), 3.58 (s, 3H, CH_3_-O), 2.49–2.46 (m, 2H, cyclohexanone C_8_-H), 2.24–2.19 (m, 5H, cyclohexanone C_6_-H, thiazole-CH_3_), 2.10 (s, 3H, DHP-CH_3_), 1.94–1.76 (2 × m, 2H, cyclohexanone C_7_-H). ^13^C-NMR (125MHz, DMSO-d_6_): 194.66, 167.97, 167.53, 152.10, 148.62, 147.43, 140.14, 138.35, 119.62, 112.12, 111.96, 110.28, 78.20, 55.88, 55.61, 53.32, 53.13, 37.32, 37.07, 26.70, 26.79, 21.35, 17.71, 17.33. (IR (KBr) ν (cm^−1^): 3229 (DHP N-H), 2937 (C-H, aromatic), 2837 (C-H, aliphatic), 1654, 1620 (C=O, amide, ketone); MS *m*/*z* (%): 437 (25) [M − 2]^+^, 324 (100), 308 (3), 280 (8). Anal. Calcd: C, 63.14; N, 9.60; O, 14.63. Found: C, 63.03; N, 9.46; O, 14. 46.

*4-(2,4-Dichlorophenyl)-2-methyl-N-(2-methylthiazol-4-yl)-5-oxo-5,6,7,8-tetrahydroquinoline-3-carboxamide* (**B1**)**:** Yield 35%; yellow powder; mp. 248 °C; ^1^H-NMR (500MHz, DMSO-*d*_6_) δ: 12.49 (s, 1H, amide N-H), 7.58 (s, 1H, C_5_-H), 7.30 (d, 2H, *J* = 8.5Hz, C_2_-H, C_3_-H), 6.76 (s, 1H, thiazole-H), 3.17 (m, 2H, cyclohexanone C_8_-H), 2.60-2.55 (m, 2H, cyclohexanone C_6_-H), 2.50 (s, 3H, thiazole-CH_3_), 2.24 (s, 3H, DHP-CH_3_), 2.08 (m, 2H, cyclohexanone C_7_-H). ^13^C-NMR (125MHz, DMSO-d_6_): 197.06, 164.86, 164.46, 158.40, 156.61, 147.39, 143.96, 135.67, 133.25, 133.09, 131.16, 128.60, 127.04, 123.70, 108.88, 57.92, 33.23, 23.31, 21.43, 19.02, 17.23. IR (KBr) ν (cm^−1^): 3179 (DHP N-H), 3062 (C-H, aromatic), 2953 (C-H, aliphatic), 1692, 1640 (C=O, amide, ketone); MS *m*/*z* (%): 447 (2) [M + 2]^+^, 445 (6) [M], 410 (50), 332 (92), 296 (19), 272 (28), 165 (18), 98 (17), Anal. Calcd: C, 56.51; N, 9.41; O, 7.17. Found: C, 56.35; N, 9.32; O, 7.01.

*4-(4-Bromophenyl)-2-methyl-N-(2-methylthiazol-4-yl)-5-oxo-5,6,7,8-tetrahydroquinoline-3-carboxamide* (**B2**): Yield 30%; yellow powder; mp. 261 °C; ^1^H-NMR (500MHz, DMSO-*d*_6_) δ: 12.42 (s, 1H, amide N-H), 7.47 (d, 2H, *J* = 7.0Hz, C_3_-H_,_ C_5_-H), 7.08 (d, 2H, *J* = 7.0Hz, C_2_-H_,_ C_6_-H), 6.76 (s, 1H, thiazole-H), 3.14 (t, 2H, *J* = 6.0Hz, cyclohexanone C_8_-H), 2.58 (t, 2H, cyclohexanone C_6_-H), 2.47 (s, 3H, thiazole-CH_3_), 2.23 (s, 3H, DHP-CH_3_), 2.11–2.08 (m, 2H, cyclohexanone C_7_-H). ^13^C-NMR (125MHz, DMSO-d_6_): 197.37, 165.07, 164.94, 157.79, 156.67, 147.34, 146.86, 136.86, 131.48, 130.80, 130.34, 124.08, 121.20, 108.80, 79.63, 33.36, 23.12, 21.39, 17.24. IR (KBr) ν (cm^−1^): 3182 (DHP N-H), 3048 (C-H, aromatic), 2947 (C-H, aliphatic), 1693, 1644 (C=O, amide, ketone); MS *m*/*z* (%): 457 (21) [M + 2]^+^, 455 (20) [M]^+^, 342 (100), 262 (7), 235 (25), 162 (16), Anal. Calcd: C, 55.27; N, 9.21; O, 7.01. Found: C, 55.17; N, 9.16; O, 6.84.

*4-(3-Chlorophenyl)-2-methyl-N-(2-methylthiazol-4-yl)-5-oxo-5,6,7,8-tetrahydroquinoline-3-carboxamide* (**B3**): Yield 25%; yellow powder; mp. 241 °C; ^1^H-NMR (500MHz, DMSO-*d*_6_) δ: 12.41 (s, 1H, amide N-H), 7.31–7.27 (m, 2H, C_4_-H_,_ C_5_-H), 7.20 (s, 1H, C_2_-H), 7.09 (d, 1H, *J* = 7.0Hz, C_6_-H), 6.75 (s, 1H, thiazole-H), 3.15 (t, 2H, *J*=6.0Hz, cyclohexanone C_8_-H), 2.67–2.51 (m, 2H, cyclohexanone C_6_-H), 2.47 (s, 3H, thiazole-CH_3_), 2.23 (s, 3H, DHP-CH_3_), 2.12–2.08 (m, 2H, cyclohexanone C_7_-H). ^13^C-NMR (125MHz, DMSO-*d*_6_): 197.26, 164.95, 157.63, 156.68, 147.31, 146.36, 139.61, 132.41, 131.50, 129.69, 128.08, 127.74, 126.80,123.92, 108.76, 56.49, 33.37, 23.11, 21.35, 19.01, 17.22. IR (KBr) ν (cm^−1^): 3154 (DHP N-H), 3062 (C-H, aromatic), 2924 (C-H, aliphatic), 1694, 1672 (C=O, amide, ketone); MS *m*/*z* (%): 413 (32) [M + 2]^+^, 411 (13) [M], 300 (58), 298 (100), 280 (12), 165 (21), Anal. Calcd: C, 61.24; N, 10.20; O, 7.77; Found: C, 61.13; N, 10.16; O, 7.56.

*2-Methyl-N-(2-methylthiazol-4-yl)-5-oxo-4-(3,4,5-trimethoxyphenyl)-5,6,7,8-tetrahydroquinoline-3-carboxamide* (**B4**): Yield 30%; yellow powder; mp. 238 °C; ^1^H-NMR (500MHz, DMSO-*d*_6_) δ: 12.39 (s, 1H, amide N-H), 6.76 (s, 1H, thiazole-H), 6.46 (s, 2H, C_2_-H_,_ C_6_-H), 3.61 (s, 6H, -OCH_3_), 3.58 (s, 3H, -OCH_3_), 3.15 (t, 2H, *J* = 6.0Hz, cyclohexanone C_8_-H), 2.67 (t, 2H, *J* = 6.0Hz, cyclohexanone C_6_-H), 2.47 (s, 3H, thiazole-CH_3_), 2.23 (s, 3H, DHP-CH_3_), 2.13–2.10 (m, 2H, cyclohexanone C_7_-H). ^13^C-NMR (125MHz, DMSO-*d*_6_): 197.29, 165.47, 164.69, 157.22, 156.88, 152.46, 147.94, 147.28, 137.13, 132.97, 131.83, 124.52, 108.60,106.26, 102.78,79.63, 60.38, 56.18, 33.36, 23.06, 21.47, 17.26. IR (KBr) ν (cm^−1^): 2994 (C-H, aromatic), 2960 (C-H, aliphatic), 1693, 1664 (C=O, amide, ketone); MS *m*/*z* (%): 467 (40) [M]^+^, 354 (100), 338 (16), 323 (21), 311 (68), Anal. Calcd: C, 61.66; N, 8.99; O, 17.11. Found: C, 61.46; N, 8.75; O, 17.07.

*4-(4-Chlorophenyl)-2-methyl-N-(2-methylthiazol-4-yl)-5-oxo-5,6,7,8-tetrahydroquinoline-3-carboxamide* (**B5**): Yield 27%; yellow powder; mp. 251 °C; ^1^H-NMR (500MHz, DMSO-d_6_) δ: 12.41 (s, 1H, amide N-H), 7.33 (d, 2H, *J* = 2.0Hz, C_3_-H, C_5_-H), 7.15 (d, 2H, *J* = 2.0Hz, C_2_-H, C_6_-H_)_, 6.75 (s, 1H, thiazole-H), 3.14 (m, 2H, cyclohexanone C_8_-H), 2.50(s, 3H, thiazole-CH_3_ overlapped with DMSO peak), 2.57 (m, 2H, cyclohexanone C_6_-H), 2.22 (s, 3H, DHP-CH_3_), 2.09 (s, m, 2H, cyclohexanone C_7_-H). ^13^C-NMR (125MHz, DMSO-d_6_): 197.36, 165.10, 164.93, 157.58, 156.68, 147.33, 146.86, 136.46, 132.53, 131.59, 130.05, 127.89, 124.11, 108.77, 30.36, 23.11, 21.39, 21.27, 17.21. (IR (KBr) ν (cm^−1^): 3079 (C-H, aromatic), 2967 (C-H, aliphatic), 1692, 1681 (C=O, amide, ketone); MS *m*/*z* (%): 413 (10) [M + 2]^+^, 411 (28) [M + 1]^+^, 300 (61), 280 (13), 268(6), 165(21), Anal. Calcd: C, 61.24; N, 10.20; O, 7.77. Found: C, 61.12; N, 10.11; O, 7.24.

*4-(3-Ethoxy-4-hydroxyphenyl)-2-methyl-N-(2-methylthiazol-4-yl)-5-oxo-5,6,7,8-tetrahydroquinoline-3-carboxamide* (**B6**): Yield 29%; yellow powder; mp. 267 °C; ^1^H-NMR (500MHz, DMSO-d_6_) δ: 11.62 (s, 1H, amide N-H), 8.55 (s, 1H, OH), 6.72 (s, 1H, C_2_-H_)_, 6.66 (s, 1H, thiazole-H), 6.59 (d, 1H, *J* = 8.0 Hz, C_5_-H), 6.52 (d, 1H, *J* = 8.0 Hz, C_6_-H_)_, 3.90-3.78 (m, 2H, CH_3_-CH_2_-O), 2.48–2.45 (m, 2H, cyclohexanone C_8_-H), 2.23 (s, 3H, C_6_-H, thiazole-CH_3_)_,_ 2.21-2.19 (m, 2H, cyclohexanone), 2.09 (s, 3H, DHP-CH_3_), 1.93–1.78 (2 × m, 2H, cyclohexanone C_7_-H), 1,23(t, 3H, *J* = 7.0, CH_3_-CH_2_-O). ^13^C-NMR (125MHz, DMSO-*d*_6_): 194.54, 152.26, 151.76, 146.45, 146.29, 140.34, 138.59, 138.10, 134.89, 118.42, 115.54, 112.32, 110.69, 108.44, 107.74, 56.02, 37.32, 36.79, 30.83, 26.78, 21.28, 17.72, 17.34. IR (KBr) ν (cm^−1^): 3320 (O-H), 3277 (C-H, aromatic), 2925 (C-H, aliphatic), 1667, 1602 (C=O, amide, ketone); MS *m*/*z* (%): 439 (5) [M + 2]^+^, 326 (8), 299 (49), 268 (23), 188 (74), 162 (100), Anal. Calcd: C, 63.00; N, 9.58; O, 14.59. Found: C, 62.87; N, 9.36; O, 14.48.

*4-(4-Methoxyphenyl)-2-methyl-N-(2-methylthiazol-4-yl)-5-oxo-5,6,7,8-tetrahydroquinoline-3-carboxamide* (B7): Yield 25%; yellow powder; mp. 166 °C; ^1^H-NMR (500MHz, DMSO-*d*_6_) δ: 12.35 (s, 1H, amide N-H), 7.05 (d, 2H, *J* = 8.0 Hz, C_5_-H, C_3_-H), 6.81 (d, 2H, *J* = 8.0Hz, C_2_-H, C_6_-H), 6.74 (s, 1H, thiazole-H), 3.71 (s, 3H, CH_3_-O),3.37(s, 3H, thiazole-CH_3_), 3.12 (t, 2H, cyclohexanone C_8_-H), 2.57 (t, 2H, cyclohexanone C_6_-H), 2.22 (s, 3H, DHP-CH_3_), 2.09-2.07 (m, 2H, cyclohexanone C_7_-H). ^13^C-NMR (125MHz, DMSO-*d*_6_): 197.44, 165.52, 164.61, 158.92, 157.27, 156.85, 148.01, 147.27, 131.96, 129.62, 124.74, 113.33, 108.64, 79.63, 55.37, 33.40, 23.07, 21.48, 19.02, 17.24. (IR (KBr) ν (cm^−1^): 3011 (C-H, aromatic), 2924 (C-H, aliphatic), 1691, 1609 (C=O, amide, ketone); MS *m*/*z* (%): 407 (4) [M]^+^, 294 (21), 408 (26), 269 (62), 252 (19), 162 (100), Anal. Calcd: C, 64.85; N, 10.31; O, 11.78. Found: C, 64.65; N, 10.41; O, 11.64.

*4-(3,4-Dimethoxyphenyl)-2-methyl-N-(5-methylthiazol-2-yl)-5-oxo-5,6,7,8-tetrahydroquinoline-3-carboxamide* (**B8**): Yield 30%; yellow powder; mp. 219 °C; ^1^H-NMR (500MHz, DMSO-*d*_6_) δ: 12.42 (s, 1H, amide N-H), 6.83 (d, 1H, *J* = 8.0Hz, C_6_-H), 6.79 (s, 1H, *J* = 8.0Hz, C_2_-H), 6.76 (s, 1H, thiazole-H), 6.64 (d, 1H, *J* = 8.0Hz, C_5_-H), 3.71 (s, 3H, CH_3_-O), 3.60 (s, 3H, CH_3_-O), 3.12 (t, 2H, cyclohexanone C_8_-H), 2.59-2.55 (m, 2H, cyclohexanone C_6_-H), 2.45 (s, 3H, thiazole-CH_3_), 2.23 (s, 3H, DHP-CH_3_), 2.09-2.11-2.08 (m, 2H, cyclohexanone C_7_-H). ^13^C-NMR (125MHz, DMSO-d_6_): 197.45, 165.58, 164.66, 157.18, 156.88, 148.50, 148.05, 147.93, 147.32, 147.25, 131.94, 129.73, 124.84, 121.10, 112.47, 111.34, 108.65, 63.26, 55.76, 33.37, 23.05, 21.49, 17.25. (IR (KBr) ν (cm^−1^): 3117 (C-H, aromatic), 2936 (C-H, aliphatic), 1667, 1604 (C=O, amide, ketone); MS *m*/*z* (%): 437 (1) [M + 2]^+^, 407 (4), 294 (18), 269 (65), 252 (15), 162 (100), Anal. Calcd: C, 63.14; N, 9.60; O, 14.63. Found: C, 62.59; N, 9.40; O, 14.82.

### 2.2. Biological Evaluation

#### 2.2.1. Cell Lines

Cell lines used for cytotoxicity assays included MCF-7 (human breast adenocarcinoma), A549 (human lung adenocarcinoma), K562 (human chronic myelogenous leukemia) and HEK-293 (human embryonic kidney) cells and were obtained from the Iranian Biological Resource Center, Tehran, Iran. MES-SA-DX5 multidrug resistant human uterine sarcoma, over-expressing P-gp and their parental non-resistant MES-SA cell lines were purchased from Sigma-Aldrich (St. Louis, MO, USA). All cells, except for HEK-293, were cultured in RPMI 1640 supplemented with 10% FBS and 100 units/mL penicillin-streptomycin at 37 °C in humidified air containing 5% CO_2_. HEK-293 cells were maintained in DMEM-F12 medium containing L-glutamine 2 mM supplemented with FBS and penicillin-streptomycin. MES-SA-DX5 cells were maintained in media containing 100 nM doxorubicin in order to ensure that only drug-resistant cells continue to grow.

#### 2.2.2. Reagents

RPMI 1640 and DMEM-F12 growth mediua, L-glutamine, penicillin and streptomycin were purchased from Biosera (Nuaille, France). Fetal bovine serum (FBS) was obtained from Gibco (Carlsbad, CA, USA). Doxorubicin, cisplatin and 3-(4,5-dimethylthiazol-2-yl)-2,5-diphenyltetrazolium bromide (MTT) were purchased from EBEWE Pharma (Unterach, Austria) and Sigma Aldrich (St. Louis, MO, USA).

#### 2.2.3. MDR Reversal Assay

Rhodamine 123 (Rh123) is a well-known substrate of P-gp and it is readily pumped out of the cells by this ABC transporter. The amount of retained Rh123 in the cells indicates the efflux function of the target transporter. Flow cytometric analysis was used for assessment of intracellular Rh123 as previously described [23].

Briefly, a suspension of the multidrug resistant uterine sarcoma MES-SA-DX5 cells at a density of 5 × 10^5^ cells/mL in growth medium was prepared. Four-hundred microliters (400 µL) of the synthesized compounds with final concentration of 5, 10, and 25 µM were added to 500 µL of the cell suspension. Verapamil was also tested as a positive control. After an incubation of 20 min, 50 µL of Rh123 at a final concentration of 5 µM was added and incubated for further 30 min at 37 °C. The cells were then centrifuged and washed twice with ice-cold PBS, and then resuspended in PBS and kept on ice. Rh123 accumulated in cells was then quantified by a FACSCalibur flow cytometer (Becton Dickinson, San Jose, CA, USA) with excitation and emission wavelengths of 488 and 530 nm, respectively. Geometric means of fluorescence intensity values were determined using Flowing Software (http://flowingsoftware.btk.fi/).

#### 2.2.4. Cell Cycle Analysis

The analysis of cells in different phases of cell cycle and sub-G1 was performed using propidium iodide (PI)-RNase assay by flow cytometry. MES-SA-DX5 cells were seeded in 12-well plates (1 × 10^6^ cells/well) and treated with different concentrations of synthesized compounds for 24 h. The cells were then collected, washed with PBS, and fixed with 70% ethanol overnight at –20 °C. After at least 24 h, fixed cells were washed with PBS and subsequently stained with a DNA staining solution containing PI 20 μg/mL and RNase 200 μg/mL at room temperature for 30 min in the dark. Twenty thousand cells of each sample were analyzed using a BD FACSCalibur flow cytometer (BD Biosciences, USA) and the amount of the cells in sub-G1, G0/G1, S, and G2/M phases were estimated using CellQuest (Becton Dickinson, San Jose, CA, USA) software.

#### 2.2.5. Cell Viability Assay

Cytotoxicity of the test compounds were evaluated by MTT assay [24]. In this method, reduction of MTT, a yellow tetrazolium salt to the purple formazan, by cellular dehydrogenase enzymes is determined as an index of cell viability. Cancer cells were seeded into 96-well plates (3–5 × 10^3^ cells/well) and incubated for 24 h at 37 °C. Afterwards, four different concentrations of synthesized derivatives were added in triplicate and the cells were incubated at 37 °C for another 72 h, after which the media was replaced with MTT solution at a concentration of 0.5 mg/mL was added for MTT assay. After another 4h, formazan crystals were dissolved in 200 μL DMSO for 90 min and absorbance was measured at a wavelength of 570 nm with background correction at 650 nm using a microplate reader (model 680, Bio-Rad, Hercules, CA, USA) and IC_50_ for each compound was calculated with CurveExpert version 1.34 for Windows.

### 2.3. Statistical Analysis

All experiments were carried out in triplicate and were repeated 3–5 times. The results are reported as mean ± S.E.M.. The differences between various treatments were analyzed by one-way ANOVA with an LSD post hoc test. Statistical analysis was performed using the SPSS program version 14.0 (Chicago, IL, USA) for Windows.

## 3. Results and Discussion

### 3.1. Synthesis

The synthetic routes for the target compounds **A1**–**A8** and **B1**–**B8** are depicted in Scheme 1 and the structures are presented in Table 1. Compound 1-((2-methylthiazol-4-yl) amino) pentane-2,4-dione was synthesized by the reaction of the commercially available 2-methylthiazol-4-amine with 2,2,6-trimethyl-4*H*-1,3-dioxin-4-one in xylene under reflux condition. According to Scheme 1, the final products were synthesized by treatment of obtained intermediate with 1,3-cyclohexadione and different appropriate arylaldehydes in the presence of excess amounts of ammonium acetate in refluxing ethanol. In order to synthesis the tetrahydroquinoline derivatives; the corresponding hexahydroquinoline compound was oxidized in the presence of MnO_2_ in ethanol under reflux condition for 24–48 h.

### 3.2. Biological Evaluations

#### 3.2.1. MDR Reversal Assay

Accumulation of Rhodamine123 (Rh123), caused by inhibition of P-gp-mediated efflux, was determined by flow cytometry in the multidrug resistant uterine sarcoma MES-SA-DX5 cell line. Verapamil was used as a positive control. MES-SA-DX5 is a resistant cell line, which overexpresses P-gp as a result of constant exposure to doxorubicin. Alterations in the amount of the fluorescent Rh123 retained inside the MES-SA-DX5 cells can be plausibly related to the inhibition of the activity of P-gp efflux pump in the cells (Figure 1). The ratio of the geometric mean value of cells treated with synthesized compounds to the geometric mean value of the control untreated cells are show in Figure 2. The findings demonstrate a clear dose-dependent effect at 5, 10, and 25 µM on Rh123 accumulation for most of the compounds. Compounds **A1**, **A2**, and **B2** showed the highest P-gp inhibitory activities.

#### 3.2.2. Effect of Synthesized Derivatives on Cell Cycle

The effect of the most potent compounds on the cell cycle distribution and apoptosis induction in MES-SA-DX5 resistant cells was evaluated using propidium iodide (PI)-RNase assay by flow cytometry. Table 2 shows the distribution of the cells in the sub-G1, G0/G1, S and G2/M phases of the cell cycle. The results indicate that the number of cells in sub-G1 phase, which shows the apoptotic cells, is significantly increased after treatment with 100 µM concentrations of **A1**, **A2**, **A3**, **A5**, **B1**, **B2**, and **B5** compounds. Therefore, it could be concluded that these compounds cause cytotoxicity in drug resistant MES-SA-DX5 cells by induction of apoptosis in these cells. Furthermore, treatment of MES-SA-DX5 cells with compounds **A1** and **B1** caused a considerable increase in G2/M and a decrease in G0/G1 phase cells. Representative histograms of the most potent derivatives are shown in Figure 3.

#### 3.2.3. Cell Viability Assay

Cell viability of different cancer cell lines exposed to synthesized derivatives was evaluated using the MTT reduction assay. The average IC_50_ values in tested cell lines are summarized in Table 3. In general, the compounds had the lowest IC_50_ values against K562 cells, which is usually expected in hematological malignancies compared to solid tumors.

Interestingly, among tested solid tumors, the synthesized derivatives **A1**, **A2**, **A5**, **B1**, and **B2** were more effective against MES-SA-DX5 cells. These findings are particularly interesting if we note that MES-SA-DX5 is a multidrug resistant cell line and any compound effective against these cancer cells could have high potential as therapeutic agents against drug resistant tumors. We also tested the more promising derivatives against parental non-resistant MES-SA cells and the IC_50_ values were invariably higher in these cells compared to MES-SA-DX5-resistant cells. It should be noted that parental MES-SA cells are more than 200 times more sensitive to doxorubicin compared to resistant cells.

Since these compounds have a moderate effect against cancer cell lines, it can be deduced that they should possess an inherent cytotoxicity. On the other hand, they also have a pronounced P-gp inhibitory effect (Figure 2). Indeed, the combination of these two biological properties may lead to an additive effect in resistant cells. In other words, the P-gp inhibitory activity contributes to the retention of the derivatives within the cell, where they can better exert their antiproliferative action.

The effect of the active compounds was also tested against HEK-293 non-cancerous cell line. Compounds **A2**, **A3**, **A5**, **B2**, **B3**, and **B5** were derivatives with IC_50_ values higher than 50 µM in these normal cells. Putting together all MTT results, it can be concluded that compounds **A2** and **B2** are the most promising agents, which show relative selectivity of action against resistant cells compared to non-resistant and also non-cancerous cell lines.

#### 3.2.4. Structure-Activity Relationships

Considering the MDR reversal and cytotoxicity results, the following structure-activity relationships could be construed for the synthesized compounds:

#### 3.2.5. Comparison between 1,4-Dihydropyridine and Pyridine Structures

The findings in MES-DX-5 drug resistant cells indicated that oxidation of 1,4-dihydropyridine ring in **A** (hexahydroquinoline) compounds could not significantly alter the MDR reversal effect of derivatives as compared with **B** (tetrahydroquinoline) series. As it can be observed in Figure 1 and Figure 2, **A** compounds show only a slight overall superiority over **B** agents. Both groups of hexahydroquinoline and tetrahydroquinoline derivatives in general demonstrated weak to moderate cytotoxic effects against MES-SA-DX5 cells in the MTT assay (Table 3).

On the other hand, oxidation of 1,4-dihydropyridine ring to pyridine counterpart resulted in an increase of the cytotoxic potential of compounds against K562 cells in most cases; e.g., **B1** (IC_50_ = 6.7 µM), **B2** (IC_50_ = 10.1 µM) and **B5** (IC_50_ = 10.4 µM) demonstrated superior activity compared to their pyridine counterparts.

#### 3.2.6. Influence of the Substituted Group at C_4_ Position of the Phenyl Ring

The MDR reversal results indicated that most of the compounds bearing halogen moieties including Cl (**A1**, **A3**, **A5**, **B1**, **B5**) and Br (**A2** and **B2**) on the phenyl ring showed superior MDR reversal effect compared to the compounds bearing other substitutes, such as methoxy, methyl, and hydroxyl, in both **A** and **B** series.

As for the cytotoxic effect, the findings showed that the best compounds especially against K562 cells were those bearing halogen moieties on the *para* position of phenyl ring (**A1**, **A2**, **B1**, **B2**) compared to compounds bearing other substitutes in both **A** and **B** series. It should also be mentioned that *meta*-chlorinated derivatives (**A3** and **B3**) demonstrated weaker cytotoxic activity in cell lines compared to *para*-chlorinated counterparts (**A5** and **B5**).

## 4. Conclusions

A series of 2-methyl-*N*-(2-methylthiazol-4-yl)-5-oxo-1,4,5,6,7,8-hexahydroquinoline-3-carboxamide derivatives (**A** series) and their tetrahydroquinoline analogs (**B** series) were synthetized and evaluated for MDR reversal and cytotoxic properties. Examination of MDR reversal in MES-SA-DX5 drug resistant cells by flow cytometric detection of Rh123 showed that compounds especially those bearing Cl and Br moieties in both hexahydroquinoline and tetrahydroquinoline subgroups are able to reverse MDR in a dose dependent manner. The flow cytometric analysis of cell cycle distribution in MES-SA-DX5 cells after treatment with some of the synthesized compounds showed that especially compounds **A1** and **A2** with 5-oxo-hexahydroquinoline structure bearing 2,4-dichlorophenyl and 4-bromophenyl moieties, respectively, and their tetrahydroquinoline counterparts **B1** and **B2** induce apoptosis in MDR cells. Compounds **A2** and **B2** did not show any effect against parental non-resistant MES-SA cells and had higher IC_50_ values against HEK-293 non-cancerous cells compared to cancer cells. The results of this report show that the synthesized derivatives deserve further studies for discovery of novel MDR reversal agents.

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
