# Peer review of "5-Oxo-hexahydroquinoline Derivatives and Their Tetrahydroquinoline Counterparts as Multidrug Resistance Reversal Agents"

_molecules, 2020, doi:10.3390/molecules25081839_

Round 1

Reviewer 1 Report

The compounds reported in this manuscript showed an interesting activity as P-gp blockers.

Nevertheless, it is not clear what their mechanism of action is as anti-cancer agents.

The cytotoxicity on a non-cancer cell line should be determined at least for the most active derivative.

The computational study not confirmed by experimental data is not very significant.

The English must be revised, see for example line 82…report synthesis and biological evaluation a set of 16 novel….should be… report the synthesis and biological evaluation of a set of 16 novel….line 260...cells by of this….

In the introduction…a new set of sixteen novel…is redundant

In scheme 1 the chemial structures are horizontally flattened.

In table 1 the two columns Ar are empty

Author Response

Reviewer 1

The compounds reported in this manuscript showed an interesting activity as P-gp blockers. Nevertheless, it is not clear what their mechanism of action is as anti-cancer agents.

Response:

Thank you for this valid comment. As expressed by the reviewer, our study is mainly focused on the evaluation of P-gp modulatory effect of synthesized derivative. However, we also evaluated the inherent cytotoxic effect of the compounds against 3 other cancer cell lines including MCF-7, A-549 and K562 cells and also in this revision an additional non-cancerous cell line HEK-293. The compounds showed a moderate cytotoxic effect in addition to their MDR reversal capacity. Moreover, other reports have also shown the anticancer activity of dihydrpyridines (Elgemeie GH  et al. Novel dihydropyridine thioglycosides and their corresponding dehydrogenated forms as potent anti-hepatocellular carcinoma agents. Nucleosides, Nucleotides and Nucleic Acids. 2018;37(4):199-216; Shi X, et al. Selective anticancer activity of the novel steroidal dihydropyridine spirooxindoles against human esophageal EC109 cells. Biomedicine & Pharmacotherapy. 2017;96:1186-92).

However, since we have not investigated other anticancer mechanisms of these compounds in depth and it has not been the main goal of the study, the word anticancer was avoided in the title, abstract and text.

The cytotoxicity on a non-cancer cell line should be determined at least for the most active derivative.

Response:

We appreciate this valid observation. As suggested, we performed additional experiments with HEK-293 non-cancerous cells and tested the 8 more promising compounds as well as standard chemotherapeutic agents, doxorubicin and cisplatin. The data are added to Table 3.  Discussion of these data were added to the manuscript (Line 497-501).

The computational study not confirmed by experimental data is not very significant.

Response:

We have done the computational study to corroborate the experimental findings. However, since this is the opinion of the reviewer that these data do not help with this purpose, we omitted the docking study from the manuscript to make it more straight forward and less speculative.

The English must be revised, see for example line 82…report synthesis and biological evaluation a set of 16 novel….should be… report the synthesis and biological evaluation of a set of 16 novel….line 260...cells by of this….

Response:

This mistake was corrected. Thank you. We also tried our best to recheck the manuscript and correct any grammatical or orthographical errors.

In the introduction…a new set of sixteen novel…is redundant

Response:

Corrected. Thank you.

In table 1 the two columns Ar are empty

Response:

This must have been a technical problem that occurred during PDF generation by the submission portal. We made some changes in the Word file and hopefully this problem will be avoided.

In scheme 1 the chemical structures are horizontally flattened.

Response:

We tried our best to draw the chemical structures according to the standard style and also similar papers published in Molecules.

Reviewer 2 Report

Synthesis of novel 1,4,5,6,7,8-hexahydro 5-oxo quinoline-3-carboxamide derivatives bearing 4-methylthiazole moiety and their tetrahydroquinoline counterparts was described in the paper. Unfortunately, the structure of the compounds MD and OMD series was proved insufficiently. The elemental analysis or HRMS data, and 13C NMR spectra for all compounds are missing. The number of the 1H for MD compounds does not match their structure. The anticancer activity was studied thoroughly by various methods. Some compounds showed good cytotoxicities against MES-SA-DX5 cells and therapeutic potential in drug resistant cancers. The paper deserves publication in Molecules after addressing the mentioned above and the following points.

  1. The last two phrases in the abstract and conclusion repeat each other.
  2. It is necessary to include spectral data for intermediate product N-(2-methylthiazol-4-yl)-3-oxobutanamide in the experimental part. Incidentally its formulae in the Scheme 1 and name in the text (line 98) are wrong.
  3. The meanings of aryl radicals in Scheme 1 and Table 1 should be included.
  4. The references should be corrected according to the journal rules.

Author Response

Reviewer 2

Synthesis of novel 1,4,5,6,7,8-hexahydro 5-oxo quinoline-3-carboxamide derivatives bearing 4-methylthiazole moiety and their tetrahydroquinoline counterparts was described in the paper. Unfortunately, the structure of the compounds MD and OMD series was proved insufficiently. The elemental analysis or HRMS data, and 13C NMR spectra for all compounds are missing. 

Response:

Thank you for this important suggestions. As required, we performed the related measurements and added the new data including elemental analysis  and 13C NMR spectra to the manuscript and the supplemental file.

The number of the 1H for MD compounds does not match their structure.

Response:

The 1H-NMR spectra are carefully checked and spectral analysis are rewritten.

The anticancer activity was studied thoroughly by various methods. Some compounds showed good cytotoxicities against MES-SA-DX5 cells and therapeutic potential in drug resistant cancers. The paper deserves publication in Molecules after addressing the mentioned above and the following points.The last two phrases in the abstract and conclusion repeat each other.

Response:

The last paragraph in conclusion section was rephrased.

It is necessary to include spectral data for intermediate product N-(2-methylthiazol-4-yl)-3-oxobutanamide in the experimental part. Incidentally its formulae in the Scheme 1 and name in the text (line 98) are wrong.

Response:

We appreciate very much the reviewer’s careful assessment of the manuscript. The name and structure of N-(2-methylthiazol-4-yl)-3-oxobutanamide are corrected and the data of characterization are included in the manuscript and also supplemental file.

The meanings of aryl radicals in Scheme 1 and Table 1 should be included.

Response:

Scheme 1 and Table 1 were corrected as suggested.

The references should be corrected according to the journal rules.

Response:

The references were revised according to the style of the Journal.

Reviewer 3 Report

Shahraki et al describes the syntheses of some dihydropyridine derivatives in an attempt to develop molecules that can inhibit P-glycoproteins and, by extension, overcome MDR in cancer cells. The current work in a continuation of their previously reported studies published in MedChemComm (2017) and Drug Des Dev Ther (2017).

  1. General spell-check and grammar throughout the manuscript.
  2. The word “novel’” should be expunged from the title and throughout the manuscript. There is nothing novel in the 5-oxo-hexahydroquinoline and tetrahydroquinoline scaffold been reported here. The authors have in fact reported several derivatives of this scaffold in their previous work (references 20 and 22).
  3. Table 1 appears to be incomplete and does not provide any useful information about the structures of the reported compounds. This should be fixed.
  4. The reviewer finds the numbering pattern of the compounds (MD1-9 and OMD1-9) unconventional and confusing. It looks like a direct numbering system taken from a lab note. Can the authors consider using a standard numeric method, i.e. from compound 1 to infinity?
  5. Structure of aryl aldehyde should be provided in scheme 1.
  6. NMR spectra (1H and 13C) of at least MD1, MD2 and OMD 1 should be provided to confirm the identity and purity of these compounds.
  7. The authors’ claim of MDR reversal solely based on limited studies with Pgp is unacceptable. A whole range of other mechanisms could be at play here [See (i) Nat Rev Cancer (2002),2, 48-48, and (ii) J Med Chem (2017), 60, 9724-9738]. Reviewer thinks a more appropriate construction will be that the compounds reverse resistance to agent A and/or B in MDR cancer cell phenotypes. This should be addressed throughout the manuscript.
  8. Accumulation of a compound within the cell might not necessarily mean the compound inhibits efflux proteins, it could simply mean that the compound is not a substrate. The authors used MDR MES-SA/DX5, which presumably overexpress Pgps, to measure this effect. Can the authors comment on what they think will happen if the same study is repeated with an isogenic cell line of moderate to nil Pgp expression?
  9. In one instance (lines 327 to 337), the authors investigate the ability of the compounds to inhibit Pgp proteins as a function of measuring accumulation within MES-SA-DX5 (consistency with MES-SA/DX5 nomenclature!!!). In another context (lines 383 to 394), the authors discuss “potent activities of MD1, MD2 and OMD1 against MES-SA-DX5” which is of course mediocre compared to the activity of doxorubicin and cisplatin. Are the compounds cytotoxic by themselves or they just inhibit efflux proteins? If cytotoxic, they are definitely not of significant therapeutic value, relative to the controls used, due to their high micromolar range potency. If they merely inhibit efflux, how was the activity quantified? Do they, for example, potentiate or restore the activity of doxorubicin, which is highly susceptible to efflux by Pgps?
  10. Cell viability was measured using MTT assay, but this does not say anything as to whether the compounds are antiproliferative or cytotoxic? Viability, I assume, was interpreted as a function of the control (cells without drugs) which will proliferate anyways after 72 h, whereas antiproliferative compounds might simply be misconstrued as reduction in cell numbers? Can the authors comment on whether the compounds are cytotoxic or just antiproliferative?
  11. Also, any information on the effects of these compounds on normal cell lines?

Author Response

Reviewer 3

Shahraki et al describes the syntheses of some dihydropyridine derivatives in an attempt to develop molecules that can inhibit P-glycoproteins and, by extension, overcome MDR in cancer cells. The current work in a continuation of their previously reported studies published in MedChemComm (2017) and Drug Des Dev Ther (2017).

  1. General spell-check and grammar throughout the manuscript.

Response:

We appreciate this suggestion. The whole manuscript was rechecked carefully and found mistakes were corrected.

  1. The word “novel’” should be expunged from the title and throughout the manuscript. There is nothing novel in the 5-oxo-hexahydroquinoline and tetrahydroquinoline scaffold been reported here. The authors have in fact reported several derivatives of this scaffold in their previous work (references 20 and 22).

Response:

Although we used to word novel intending to express that the individual chemical structures are novel, however, this is a valid point that the reader may understand that we are claiming the whole scaffold to be novel. As suggested, we omitted the word novel from Title, Abstract and the text.

  1. Table 1 appears to be incomplete and does not provide any useful information about the structures of the reported compounds. This should be fixed.

Response:

This must have been a technical problem that occurred during PDF generation by the submission portal. We made some changes in the Word file and hopefully this problem will be avoided.

  1. The reviewer finds the numbering pattern of the compounds (MD1-9 and OMD1-9) unconventional and confusing. It looks like a direct numbering system taken from a lab note. Can the authors consider using a standard numeric method, i.e. from compound 1 to infinity?

Response:

This is a very valid suggestion for which we are very thankful. The name of the compounds were converted to (A1-A8) and (B1-B8) for MD and OMD series, respectively, throughout the whole manuscript.  

  1. Structure of aryl aldehyde should be provided in scheme 1.

Response:

The structures are provided as correctly suggested.

  1. NMR spectra (1H and 13C) of at least MD1, MD2 and OMD 1 should be provided to confirm the identity and purity of these compounds.

Response:

All 1H and 13C spectra are included in supplementary file.

  1. The authors’ claim of MDR reversal solely based on limited studies with Pgp is unacceptable. A whole range of other mechanisms could be at play here [See (i) Nat Rev Cancer (2002),2, 48-48, and (ii) J Med Chem (2017), 60, 9724-9738]. Reviewer thinks a more appropriate construction will be that the compounds reverse resistance to agent A and/or B in MDR cancer cell phenotypes. This should be addressed throughout the manuscript.

Response:

As correctly mentioned by the reviewer, MDR has several different mechanisms, one of which is P-gp modulation. In other words, although MDR has several different mechanisms, one of the most established ones is definitely P-gp modulation. It is also mentioned by the reviewer in the first paragraph that we have used MDR reversal phrase by extension to P-gp inhibitory effect.

In this study we measure a well-known substrate of P-gp in a cell line that overexpresses P-gp, and hence can conclude that these compounds are really modulating the function of P-gp.

Other studies that have measured P-gp modulation has similarly expressed this effect as MDR reversal (Teng YN et al., Caffeic Acid Attenuates Multi-Drug Resistance in Cancer Cells by Inhibiting Efflux Function of Human P-glycoprotein. Molecules. 2020 25(2). pii: E247; Chen HJ et al. Taxifolin Resensitizes Multidrug Resistance Cancer Cells via Uncompetitive Inhibition of P-Glycoprotein Function. Molecules. 2018 Nov 22;23(12). pii: E3055).

  1. Accumulation of a compound within the cell might not necessarily mean the compound inhibits efflux proteins, it could simply mean that the compound is not a substrate. The authors used MDR MES-SA/DX5, which presumably overexpress Pgps, to measure this effect. Can the authors comment on what they think will happen if the same study is repeated with an isogenic cell line of moderate to nil Pgp expression?

Response:

We agree with the reviewer that the accumulation of a compound within an P-gp overexpressing cell might be due to the fact that the compound is not a substrate. Indeed, we did not measure the accumulation of the synthesized derivatives themselves, but measured the accumulation of a very well-known substrate, i.e., rhodamine123, which is very effectively pumped out of the cells by P-gp.

We have shown in our previous studies by Western blot that MES-SA-DX5 cells express large quantities of P-gp (Shekari F, et al. Cytotoxic and multidrug resistance reversal activities of novel 1, 4-dihydropyridines against human cancer cells. Eur J Pharmacol. 2015;746:233-44.)

Our FACS analysis clearly shows that the synthesized derivatives cause Rhoddamine123 accumulation, hence we can safely conclude that the derivatives inhibit the efflux pump function of P-gp transporter.

As validly suggested by the reviewer, we did additional experiments on parental non-resistant MES-SA cells and the findings are reported in Table 3. As it can be seen from the data, the effect of derivatives invariably is lower against non-resistant cells. The discussion of these new data is added to the text (Lines: 476-480).

  1. In one instance (lines 327 to 337), the authors investigate the ability of the compounds to inhibit Pgp proteins as a function of measuring accumulation within MES-SA-DX5 (consistency with MES-SA/DX5 nomenclature!!!).

Response:

Thank you for the observation on MES-SA-DX5 nomenclature consistency. It was considered throughout the manuscript.

In another context (lines 383 to 394), the authors discuss “potent activities of MD1, MD2 and OMD1 against MES-SA-DX5” which is of course mediocre compared to the activity of doxorubicin and cisplatin. Are the compounds cytotoxic by themselves or they just inhibit efflux proteins? If cytotoxic, they are definitely not of significant therapeutic value, relative to the controls used, due to their high micromolar range potency. If they merely inhibit efflux, how was the activity quantified? Do they, for example, potentiate or restore the activity of doxorubicin, which is highly susceptible to efflux by Pgps?

Response:

We appreciate these valid observations. We corrected the section mentioned by the reviewer and stated that the compounds have a moderate direct effect against MES-SA-DX5 cells.

We assume that that these compounds possess both cytotoxic and P-gp inhibitory effect. However, since P-gp modulation has been the main purpose of the study and also the more pronounced biological effect of the synthesized derivatives, we focused mainly on P-gp and MDR reversal effect in the discussion of the findings thorough the text and avoided emphasizing on the anticancer effect.

These additional explanations were also added to the text (Lines 481-485):

Since these compounds have a moderate effect against cancer cell lines, it can be deduced that they should possess an inherent cytotoxicity. On the other hand, they also have a pronounced P-gp inhibitory effect (Figure 2). Indeed, the combination of these two biological properties may lead to an additive effect in resistant cells. In other words, the P-gp inhibitory activity contributes to the retention of the derivatives within the cell, where they can better exert their antiproliferative action.

  1. Cell viability was measured using MTT assay, but this does not say anything as to whether the compounds are antiproliferative or cytotoxic? Viability, I assume, was interpreted as a function of the control (cells without drugs) which will proliferate anyways after 72 h, whereas antiproliferative compounds might simply be misconstrued as reduction in cell numbers? Can the authors comment on whether the compounds are cytotoxic or just antiproliferative?

Response:

We agree with the reviewer that MTT assay cannot distinguish between antiproliferative or cytotoxic effects. It needs to be mentioned that these two expressions are sometimes used interchangeably. Some authors have interpreted the MTT results as cytotoxic effect (Ma L et al. Novel Steroidal 5α,8α-Endoperoxide Derivatives with Semicarbazone/Thiosemicarbazone Side-chain as Apoptotic Inducers through an Intrinsic Apoptosis Pathway: Design, Synthesis and Biological Studies. Molecules. 2020 Mar 7;25(5). pii: E1209).

It needs to be mentioned that since we have measured cell cycle alterations and observed that the compounds do not alter cell cycle distribution, but instead induce apoptosis, we can assume that the test compounds are cytotoxic rather than antiproliferative. Antiproliferative agents are expected to block cell cycle at a certain point.

  1. Also, any information on the effects of these compounds on normal cell lines?

Response:

We are thankful for this important suggestion. As suggested, we tested the effect of the active compounds against non-cancerous cell lines HEK-293 cells and added the data to the Table 3. The discussion of these new data are added to the manuscript (lines 497-501).

Round 2

Reviewer 1 Report

The article is suitable for the publication in Molecules

Author Response

Thank you very much.

Reviewer 2 Report

The authors have corrected the paper as requested, and the paper can be recommended for publication in Molecules.

Author Response

Thank you very much.

Reviewer 3 Report

The authors addressed most of my observations and concerns.

However, my major unresolved concern is the NMR spectra provided that neither confirmed the identity of reported compounds nor demonstrated purity. Integration was selectively done to show whatever number of protons the authors want to show and there were many protons unaccounted for. In most cases, the number of protons exceeded the expected number of protons for the reported molecules. Carbon NMR signals of the compounds were so low that they do not provide any meaningful information. Any credible medicinal chemist reviewing this paper will find this very troubling. The mass spec data (low resolution) provided also do not help as there are multiple peaks of similar intensities that suggest presence of impurities. I am unconvinced that the reported elemental analysis is consistent with the NMR/mass spec data presented. Clearly, these data will not fly in any medicinal chemistry journal. The authors recorded proton NMR at 500 MHz and also recorded carbon-13 at 500 MHz? This cannot be true! Maybe they meant carbon at 125 MHz??

At the very least, the authors should correctly identify and assign the integrated protons with their corresponding carbons, and include this data in the manuscript.

Author Response

Response:

We appreciate the careful assessment of the manuscript and acute observations of the reviewer.

As a general remark in response to the comments, we should mention that all synthesized compounds were rigorously purified in several chromatographic steps. The quality of the NMR spectra have unfortunately suffered from the presence of impurities in the solvents available to us, which have produced tiny picks in certain occasions. We apologize for this inconvenience, but they could not have been avoided due to the suboptimal quality of the used solvents. However, in addition to the fact that the extra peaks can be attributed to the solvent impurities due to their minute size, we believe that our straight forward synthetic procedure is expected to give rise to certain predictable chemical structures, which have been very carefully purified and characterized. Hence we can express sufficient confidence on the chemical characterizations of reported compounds as detailed below:

1H NMR spectra:

Considering the valuable comments about the extra peaks observed in some cases (A2, A4, B4), it should be mentioned that we reanalyzed the integration of all peaks in the mentioned cases. For instance, the integration of some minor peaks in the 1H -NMR of compound A2 are shown in the attached figure and described as follows:

The corresponding integrals for peaks observed at 2.47, 2.48 and 4.53 ppm were 0.04, 0.04 and 0.14, respectively. Needless to say that these minute peaks can be attributed to the trace impurities of solvents remained during several steps of purification as mentioned above.

As correctly suggested, all 1H NMR spectra were further carefully reanalyzed and integrated protons were assigned carefully in each case. For further clarifications, each carbon was numbered in Table 1 in order to be precisely used in the assignment of integrated protons. These changes were applied to the revised manuscript and highlighted.

13C-NMR spectra:

We are very grateful for the precise observation on the report of 13C-NMR spectra. The 13C-NMR spectra were all recorded at 125 MHz as correctly pointed out by the reviewer. This statement was corrected in the manuscript and highlighted.

Mass spectra:

Regarding the mass spectra, although we believe that most of the spectra possess sufficient clarity, we should also add that the mass spectra were monitored for primary confirmation of compound formation at earlier steps, and occasionally further purifications were also applied when necessary.

We hope that the above explanations will be sufficient to clarify the ambiguities and the revised manuscript will be acceptable for publication with the applied modifications.
